# ADVERSARIAL DISTRIBUTIONS AGAINST OUT-OF-DISTRIBUTION DETECTORS

## ABSTRACT

Out-of-distribution (OOD) detection is the task of determining whether an input lies outside the training data distribution. As an outlier may deviate from the training distribution in unexpected ways, an ideal OOD detector should be able to detect all types of outliers. However, current evaluation protocols test a detector over OOD datasets that cover only a small fraction of all possible outliers, leading to overly optimistic views of OOD detector performance. In this paper, we propose a novel evaluation framework for OOD detection that tests a detector over a larger, unexplored space of outliers. In our framework, a detector is evaluated with samples from its **adversarial distribution**, which generates diverse outlier samples that are likely to be misclassified as in-distribution by the detector. Using adversarial distributions, we investigate OOD detectors with reported near-perfect performance on standard benchmarks like CIFAR-10 vs SVHN. Our methods discover a wide range of samples that are obviously outlier but recognized as in-distribution by the detectors, indicating that current state-of-the-art detectors are not as perfect as they seem on existing benchmarks.

## 1 INTRODUCTION

Identifying whether an input datum lies outside the training data distribution is one of the canonical problems in machine learning. Over its long history, the problem has been called by multiple names, including novelty detection (Markou & Singh, 2003), outlier detection (Hawkins, 1980), one-class classification (Japkowicz et al., 1995), and more recently, out-of-distribution (OOD) detection (Hendrycks & Gimpel, 2016). Investigation of the problem has resulted in a number of real-world applications, for example, medical diagnosis (Li et al., 2019) and inspection of defective parts and products (Bergmann et al., 2019). The relevance of OOD detection is growing beyond these applications, as an OOD detector is considered an essential component of a trustworthy machine learning system. For example, without a reliable OOD detector, an image classifier trained to classify cats and dogs may incorrectly classify a human as belonging to one of these two classes (Hendrycks et al., 2019b). In order to advance the development of reliable OOD detection algorithms, a more comprehensive evaluation protocol is needed.

Current evaluation protocols adopted by the community provide a distorted view of a detector's performance for two reasons. First, OOD detectors are tested over a small fraction of possible outliers. Detectors are typically evaluated using test OOD datasets chosen by a researcher. Since the test OOD datasets do not cover the entire space of outliers, there may exist untested outliers that the detector fails to classify correctly, even though the detector perfectly detects the chosen test OOD points as shown in Figure 1. To assess a detector's performance in a more comprehensive and systematic way, a method is needed to test a detector over a larger, unexplored space beyond what is covered by the test OOD datasets.

Second, the current evaluation protocol neglects the *worst-case* behavior of an OOD detector. The average performance metric is often not sufficient to build trust on a detector, because in safety-critical applications even a single mistake can result in fatal consequences. A detector should be tested *adversarially*, through an active search for its worst-case failure mode, i.e., an outlier that is classified with maximal confidence as an inlier. Such failure cases may reveal weaknesses of the tested detector, and provide informative clues for building a better detection algorithm.

In this paper, we propose a novel evaluation protocol of OOD detectors that addresses the above-mentioned limitations of current evaluation methods. We first formulate the notion of **adversarial search** against an OOD detector, a search problem of finding an outlier that a detector classifies as an inlier with the greatest confidence. A detector may have more than one significant failure mode, and it is more informative to find a set of diverse failure cases instead of the most critical one. To that end, we propose the **adversarial distribution** against an OOD detector, which generates outlier samples that are likely to be misclassified by the given detector. Measuring OOD detection performance against samples from a detector's adversarial distribution gives an accurate and finer-grained assessment on its performance.

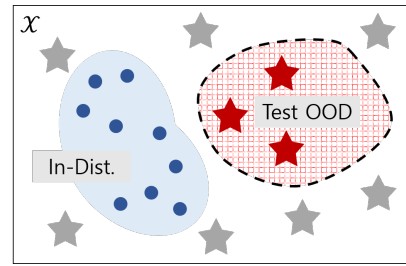

Figure 1: An illustration of OOD detection. The test data (red stars) are perfectly predicted as OOD (red hatching), but other untested outliers (gray stars) are misclassified as in-distribution. The blue shade indicates the support of the training data distribution, the ideal decision boundary for OOD detection.

To ensure that samples from an adversarial distribution are indeed outliers, the distribution needs to be supported on a **zero-inlier space**, a set without any overlap to the inlier distribution. Meanwhile, the zero-inlier space should be large so that we may observe meaningful failure mode of OOD detectors within the space. However, finding such a space is generally challenging, as the true boundary between inliers and outliers is unknown. We circumvent this challenge by building a generative model over known outliers. Our construction of zero-inlier spaces contain diverse samples.

We implement 11 previously proposed OOD detectors and investigate their behavior using their adversarial distributions. Among the tested detectors, 8 detectors report near-perfect OOD detection performance on a popular benchmark, CIFAR-10 (in) vs SVHN (out), effectively being indistinguishable with respect to their performance. Our investigation reveals that the 8 detectors in fact have diverging degrees of detection quality outside the SVHN test set. Our methods also lead to several interesting insights that suggest techniques for improving OOD detection.

Our main contributions can be summarized as follows:

- We propose the adversarial search and adversarial distributions, which can be used to evaluate OOD detection algorithms beyond a pre-defined test OOD dataset;
- We provide practical techniques to define the space of outliers that contains samples outside the test OOD dataset;
- By examining the state-of-the-art OOD detectors, we show that OOD detectors with seemingly equivalent performance differ significantly in their failure modes;

**Related Work** Testing the worst-case performance of an algorithm is highly related to evaluating adversarial robustness of the detector. Detection of adversarially perturbed outliers is investigated in previous literature where perturbation is assumed to be restricted in a small norm-ball Hein et al. (2019); Meinke & Hein (2020); Bitterwolf et al. (2020). The idea of using an autoencoder during an adversarial attack is investigated in Tu et al. (2019). The idea of generating outliers are investigated in the context of improving OOD detection Chen et al. (2020) or adversarial attack Song et al. (2018).

In Section 2, we provide essential preliminaries. Section 3 formulates adversarial search and adversarial distributions, and Section 4 introduces techniques to construct a zero-inlier space. Our main experimental results are presented in Section 5, with deeper discussions provided in Section 6. Section 7 concludes the paper.

## 2 BACKGROUND: OUT-OF-DISTRIBUTION DETECTION

### 2.1 DEFINITION

We consider a probability distribution of interest $P_{in}$, samples from which we consider as inliers. Each sample is represented as a $D$-dimensional real-valued vector. The probability density function of $P_{in}$ is denoted as $p_{in}(\mathbf{x})$. We write the support of $P_{in}$ as $\mathcal{S}_{in} = \{\mathbf{x}|p_{in}(\mathbf{x}) > 0\} \subset \mathcal{X} \subset \mathbb{R}^D$,

where $\mathcal{X}$ is the data space, the set of all possible values for data, which we assume to be compact. Then, we define OOD-ness as follows:

**Definition 1.** A vector $\mathbf{x}$ is **out-of-distribution (OOD)**, or an outlier, with respect to $P_{in}$ if $\mathbf{x}$ does not belong to the support of $P_{in}$, i.e., $\mathbf{x} \notin \mathcal{S}_{in}$. Conversely, $\mathbf{x}$ is **in-distribution** when $\mathbf{x} \in \mathcal{S}_{in}$. A distribution $Q$ having the support $\mathcal{S}_Q$ **is OOD to** $P_{in}$, when $\mathcal{S}_Q \cap \mathcal{S}_{in} = \varnothing$.

Another popular definition of OOD-ness characterizes a vector $\mathbf{x}$ as OOD when the vector belongs to a density sub-level set, $\mathbf{x} \in \{\mathbf{x}|p_{in}(\mathbf{x}) \leq \eta\}$ (Steinwart et al., 2005). However, the density sub-level set does not provide a consistent characterization of the OOD-ness. A vector classified as OOD in one coordinate may not be classified as OOD in a different coordinate, because a probability density function can be arbitrarily distorted via an invertible coordinate transform as pointed out in (Lan & Dinh, 2020). On the contrary, our definition of OOD using the density support provides an invariant characterization of outliers with respect to such transforms.

An **OOD detector** $f(\mathbf{x}) : \mathbb{R}^D \to \mathbb{R}$ is a function which outputs a larger value for an input more likely to be an outlier. A test vector $\mathbf{x}^*$ is classified as OOD $f(\mathbf{x}^*) > \eta_f$ for the threshold $\eta_f$. A **detector score** refers to the function value of $f(\mathbf{x})$. We shall assume $f(\mathbf{x})$ is bounded. In this paper, we consider a *black-box* setting, where any information other than the function value of $f(\mathbf{x})$, such as its gradient, is not accessible. An OOD detector is normally trained using an **in-distribution dataset** $\mathcal{D}_{in}$, a set of iid samples from $P_{in}$. However, some OOD detectors utilize additional datasets other than $\mathcal{D}_{in}$.

## 2.2 Evaluation of OOD Detectors

The currently accepted evaluation protocol for OOD detectors relies on one or multiple **test OOD datasets** $\mathcal{D}_{out}$, which contain a finite number of samples considered as OOD with respect to $P_{in}$ by human prior knowledge. When $P_{in}$ is a distribution of images, test OOD sets are often chosen from separately published image datasets of which contents are different from that of $P_{in}$. For example, when an image dataset of animals is selected as in-distribution, a set of digit images can be used as a test OOD dataset.

Given a test OOD dataset, an OOD detector $f(\mathbf{x})$ performs the binary classification against in-distribution dataset, and the quality of the classification is considered as an indicator for how good $f(\mathbf{x})$ is as an OOD detector. The classification result is summarized using metrics such as the area under the receiver operating characteristic curve (AUROC or AUC). AUC is a preferred metric in a number of literature, as it does not require the specification of the threshold $\eta_f$. AUC score of 1.0 indicates the perfect classification, and AUC of 0.5 implies the random guess.

The research community has been focusing on a few representative in-distribution and OOD dataset pairs, such as CIFAR-10 (in) vs SVHN (out) and Fashion-MNIST (in) vs MNIST (out). These dataset pairs become popular after the reports showing that OOD detectors built upon deep generative models, such as PixelCNN++ (Salimans et al., 2017) or Glow (Kingma & Dhariwal, 2018), fail to detect outliers (Hendrycks et al., 2019a; Nalisnick et al., 2019). In fact, the generative-model-based detectors score AUC lower than 0.5, meaning that SVHN images are more strongly perceived as CIFAR-10 than the actual CIFAR-10 images by the detectors. This observation spurred intense research efforts, and now there are multiple OOD detectors achieving AUC scores higher than 0.9 or even higher than 0.99 on CIFAR-10 vs SVHN as listed in Section 5. Given the near-perfect detection score, we question whether the detectors are indeed good OOD detectors beyond the tested examples.

## 3 Adversarial Distributions

Our goal is to test how robust the state-of-the-art OOD detectors are. The robustness of an OOD detector can be investigated through a search process to find its failure modes. We are particularly interested in failure cases where outlier examples misclassified as in-distribution by the detector. Ideally, we would search for failure modes over all possible outliers, but the true set of outliers $\mathcal{S}_{out} \equiv \mathcal{X} - \mathcal{S}_{in}$ is unknown. Instead, we specify a search space $\mathcal{T}$ which is a subset of $\mathcal{S}_{out}$ and only contains outliers. We call $\mathcal{T}$ a **zero-inlier space**. In this section, we formulate two search problems on $\mathcal{T}$. The first formulation, **Adversarial Search**, is based on optimization, and the second

formulation, **Adversarial Distribution**, is based on sampling. Details on selecting meaningful $\mathcal{T}$ will be discussed in Section 4.

## 3.1 ADVERSARIAL SEARCH

Finding failure modes of an OOD detector can be formulated as an optimization problem. For a detector $f(\mathbf{x})$ and a search space $\mathcal{T}$, an adversarial search problem is finding an element $\mathbf{x}^*$ in $\mathcal{T}$ which has the lowest detector score.

$$\mathbf{x}^* = \arg\min_{\mathbf{x} \in \mathcal{T}} f(\mathbf{x}) \quad \text{such that} \quad \mathbf{x} \in \mathcal{T} \subset \mathcal{S}_{out}, \tag{1}$$

where $\mathbf{x}^*$ is the worst-case outlier we search for. If the detector score for the worst-case outlier $f(\mathbf{x}^*)$ is greater than the score of any samples in in-distribution set, then the detector $f(\mathbf{x})$ can be confirmed as robust in the search space $\mathcal{T}$. Otherwise, we conclude that the tested detector has at least one failure mode and deviates from the ideal detector. The degree of deviation may be used to quantify the robustness of a detector.

**Example: Adversarial Attack** An example of the adversarial search is applying an adversarial perturbation on outliers so that they have small $f(\mathbf{x})$ (Bitterwolf et al., 2020). In this case, $\mathcal{T}$ is a ball centered at an outlier from a test OOD dataset $\mathcal{T} = \{\mathbf{x}| \|\mathbf{x} - \tilde{\mathbf{x}}\| \leq r, \tilde{\mathbf{x}} \in \mathcal{D}_{out}\}$. A point with the smallest detector score is found via a gradient descent method. The AUC score for classifying the perturbed samples from inliers is called Adversarial AUC (Bitterwolf et al., 2020).

## 3.2 ADVERSARIAL DISTRIBUTIONS

When the search space $\mathcal{T}$ is larger than a ball centered at an outlier, $f(\mathbf{x})$ may have multiple local minima, and each minimum may reveal different failure modes of a detector. To explore multiple failure modes in $f(\mathbf{x})$, we propose a novel approach. We deliberately design a probability distribution $p_f(\mathbf{x})$ so that samples from $p_f(\mathbf{x})$ cover multiple minima of $f(\mathbf{x})$ in $\mathcal{T}$. We call $p_f(\mathbf{x})$ an **Adversarial Distribution** of a detector $f(\mathbf{x})$.

An adversarial distribution is a Gibbs distribution, also known as an energy-based model. Its energy function is the detector score $f(\mathbf{x})$, i.e., $p_f(\mathbf{x})$ is proportional to $\exp(-f(\mathbf{x})/T)$. The support of $p_f(\mathbf{x})$ is the search space $\mathcal{T}$.

$$p_f(\mathbf{x}) = \frac{1}{Z} \exp(-f(\mathbf{x})/T) \quad \text{if } \mathbf{x} \in \mathcal{T} \subset \mathcal{S}_{out} \text{ and } p_f(\mathbf{x}) = 0 \text{ otherwise}, \tag{2}$$

where $Z = \int_{\mathcal{T}} \exp(-f(\mathbf{x})/T)d\mathbf{x}$ is the normalization constant, and $T > 0$ is called temperature. $Z < \infty$, as we have assumed the bounded $f(\mathbf{x})$ and the compact data space in Section 2.

An adversarial distribution $p_f(\mathbf{x})$ has three important properties by construction. First, samples from $p_f(\mathbf{x})$ are guaranteed to be OOD, as the support is $\mathcal{T}$. Second, an adversarial distribution assigns a high probability density on samples likely to be classified as in-distribution by the detector, i.e., $p_f(\mathbf{x})$ is high when $f(\mathbf{x})$ is small. Third, for the limit $T \to 0$, sampling from an adversarial distribution becomes equivalent to the adversarial search, as the probability mass is concentrated on $\mathbf{x}^*$ in Eq. (1). Hence, an adversarial distribution can be viewed as a relaxation of the adversarial search, and $T$ governs the degree of relaxation.

Samples from $p_f(\mathbf{x})$ can be drawn using Markov Chain Monte Carlo (MCMC). MCMC is guaranteed to visit all modes of a probability distribution at least in theory, and therefore every failure mode of a detector can be generated. Measuring how well $f(\mathbf{x})$ classifies inliers from $p_f(\mathbf{x})$'s samples using metrics such as AUC gives a comprehensive measure on the performance of the detector.

## 4 ZERO-INLIER SPACE

The support $\mathcal{T}$ of an adversarial distribution defines the search space where the robustness of an OOD detector is evaluated. While there is some degree of freedom in the choice of $\mathcal{T}$, one condition should be strictly satisfied: $\mathcal{T}$ should not contain inlier data points, i.e., should be a zero-inlier space. However, as the true decision boundary between outliers and inliers is unknown, it is challenging to ensure $\mathcal{T}$ only contains outliers.

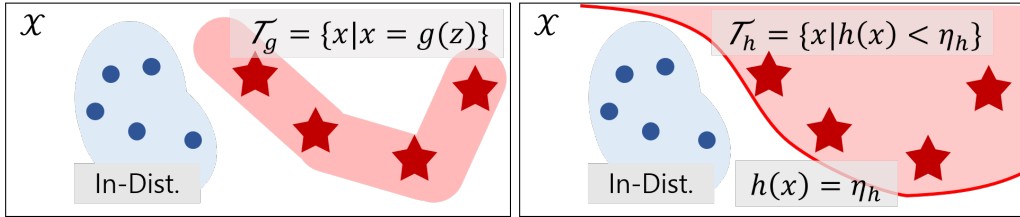

Figure 2: The illustration of $\mathcal{T}_g$ and $\mathcal{T}_h$.

In the example of the adversarial attack provided in Section 3.1, $\mathcal{T}$ is set as a ball around a known outlier. $\mathcal{T}$ is ensured to be a zero-inlier space as the radius of the ball is very small. However, such a small ball only contains examples visually indistinguishable from its center, lacking diversity. In what follows, we provide an example construction of $\mathcal{T}$ which contains a more diverse set of outliers.

Here, we describe a procedure for constructing a zero-inlier search space $\mathcal{T}$ given a set of in-distribution dataset $\mathcal{D}_{in}$ and OOD dataset $\mathcal{D}_{out}$. Note that $\mathcal{T}$ is constructed during the *evaluation* stage of OOD detectors and does not affect an OOD detector being tested. Also, the construction of $\mathcal{T}$ requires no additional data compared to a typical evaluation protocol which assumes a human-curated OOD dataset.

Instead of a ball around each outlier, we consider a generator function $g(\mathbf{z})$ spanning a set of outlier samples $\mathcal{D}_{out}$. Such a generator is able to produce an unseen sample which resembles the known outliers. As the generator is trained only using $\mathcal{D}_{out}$, it is unlikely for an inlier to be generated from $g(\mathbf{z})$. Nevertheless, to avoid a fortuitous generation of an inlier, we introduce a binary classifier $h(\mathbf{x})$ which is supervised to discriminate $\mathcal{D}_{in}$ and $\mathcal{D}_{out}$.

Let $\mathcal{T}_g$ be a set of samples generated by $g(\mathbf{z})$, and $\mathcal{T}_h$ be a set of samples classified as they are from $\mathcal{D}_{out}$. Our search space $\mathcal{T}$ is given as an intersection of $\mathcal{T}_g$ and $\mathcal{T}_h$:

$$\mathcal{T} = \mathcal{T}_g \cap \mathcal{T}_h. \tag{3}$$

## 4.1 OUTLIER-SPANNING GENERATOR

A generator $g(\mathbf{z}) : \mathcal{Z} \to \mathcal{X}$ is a map from a lower-dimensional space $\mathcal{Z} \subset \mathbb{R}^{D_Z}$ to the data space $\mathcal{X}$, where $D_Z$ smaller than the dimensionality of $\mathcal{X}$. We denote $\mathcal{T}_g$ as a set of samples that can be produced by $g(\mathbf{z})$:

$$\mathcal{T}_g = \{\mathbf{x} = g(\mathbf{z}) | \mathbf{z} \in \mathcal{Z}\}. \tag{4}$$

In our implementation of adversarial distributions, we train an autoencoder to reconstruct $\mathcal{D}_{out}$ and use its decoder as $g(\mathbf{z})$. As the autoencoder can reconstruct samples in $\mathcal{D}_{out}$ (with some error), the generator (approximately) spans the space where outliers reside. Similarly, a generator of a generative adversarial network can also serve as $g(\mathbf{z})$.

It is reported that an autoencoder may reconstruct an input vector that is significantly different from its training data even though the autoencoder is not the identity mapping (Tong et al., 2019; Gong et al., 2019; Yoon et al., 2021). This phenomenon implies that $g(\mathbf{z})$ may be capable of generating samples that is significantly different from $\mathcal{D}_{out}$. The ability of $g(\mathbf{z})$ to reach outside of $\mathcal{D}_{out}$ may positively affect the diversity of $\mathcal{T}_g$, but increases the chance of generating $\mathbf{x}$ that is close to inliers. To ensure that $\mathcal{T}_g$ only contains outliers, we introduce an additional component can filter out accidentally generated inliers.

## 4.2 BINARY CLASSIFIER SEPARATING INLIERS AND OUTLIERS

Here, we develop an additional mechanism for ensuring that no inlier is present in our search space by exploiting the fact that there is no overlap of the supports between an inlier distribution, and a distribution generating an OOD dataset. When two distributions do not overlap, there exists a binary classifier that perfectly classifies the two distributions.

**Proposition 1** (Binary classification of disjoint distributions)**.** Suppose two probability distributions $P$ and $Q$ with their supports $\mathcal{S}_P$ and $\mathcal{S}_Q$, respectively. $P$ and $Q$ are OOD to each other, if and only

if there exists a binary classifier $h^*(\mathbf{x}) : \mathcal{X} \to \mathbb{R}$ that perfectly discriminates $P$ and $Q$, i.e., there exists a threshold $\eta_{h^*} \in \mathbb{R}$ such that the classification boundary $h^*(\mathbf{x}) = \eta_{h^*}$ separates $\mathcal{S}_P$ and $\mathcal{S}_Q$.

More formal arguments will be provided in Appendix A. Assume an oracle binary classifier $h^*(\mathbf{x})$ which perfectly discriminating $\mathcal{D}_{in}$ and $\mathcal{D}_{out}$. From Proposition 1, the oracle classifier partitions the whole space into two, where one partition contains only $\mathcal{D}_{in}$ and the other contains only $\mathcal{D}_{out}$. The partition containing $\mathcal{D}_{out}$ can be chosen as a zero-inlier space.

Proposition 1 holds approximately for an empirically obtained classifier $h(\mathbf{x})$. A binary classifier often achieves AUC score higher than 0.9999 when evaluated on the test splits of $\mathcal{D}_{in}$ and $\mathcal{D}_{out}$. Hence, we define an outlier-side of $h(\mathbf{x})$'s decision boundary as $\mathcal{T}_h$. Assuming that $h(\mathbf{x})$ assigns lower value for outliers,

$$\mathcal{T}_h = \{\mathbf{x}|h(\mathbf{x}) < \eta_h,\ \mathbf{x} \in \mathcal{X}\}, \tag{5}$$

for a threshold $\eta_h$. However, a real-world classifier is not an oracle $h(\mathbf{x}) \neq h^*(\mathbf{x})$ and can never be perfect. The classifier may misclassify or may be vulnerable to adversarial perturbations. To minimize the effect, we apply a few techniques. First, we set the threshold $\eta_h$ lower than a **zero-inlier threshold**, which is the smallest $h(\mathbf{x})$ value among all training, validation, and testing inlier examples. Second, test-time augmentation is applied on $h(\mathbf{x})$ to make the prediction more robust. We use horizontal flip and rotations of 90, 180, 270 degrees. Total 5 predictions are averaged. Detailed discussion on the robustness of $h(\mathbf{x})$ is given in Section E.1.

## 5 EXPERIMENTS

In our experiments, we aim to reveal previously unknown failure modes of the state-of-the-art OOD detectors using adversarial distributions. We first show that adversarial distributions can find the known weaknesses of baseline detectors. Then, we apply our methods on the state-of-the-art OOD detectors which show near-perfect performance on a popular benchmark. Finally, we show that the failure modes of the detectors are not readily transferable. However, a simple ensemble does not improve the robustness of OOD detection significantly. Table 1 summarizes our experimental results.

### 5.1 EXPERIMENTAL SETTINGS

**Datasets**   We use CIFAR-10 as in-distribution data $\mathcal{D}_{in}$. SVHN and CelebA are used as test OOD datasets $\mathcal{D}_{out}$. All datasets are splitted into training, validation, and testing sets. All data are $32 \times 32$ RGB images. Details on datasets can be found in Appendix B.

**OOD Detectors**   We implement 11 previously proposed outlier detectors. The detectors are grouped into two, the weak and the strong, based on their performance on CIFAR-10 vs SVHN. The *weak* detectors are outlier detectors that fail dramatically on the benchmark by predicting images in SVHN being more likely to be in-distribution than images in CIFAR-10.

The following is the list of the weak detectors used in our experiment.

- **Autoencoder** (AE) (Rumelhart et al., 1986): A neural network trained to reconstruct its input. The detector score $f(\mathbf{x})$ is the reconstruction error of $\mathbf{x}$ (Japkowicz et al., 1995).
- **PixelCNN++** (PXCNN) (Salimans et al., 2017): An autoregressive generative model for images.
- **Glow** (Kingma & Dhariwal, 2018): A generative model based on a normalizing flow.

For PXCNN and Glow, the negative log-likelihood is used as the detector score.

We select *strong* detectors satisfying three criteria: 1) AUC score on CIFAR-10 vs SVHN is higher than 0.9. 2) The code is publicly available and written in PyTorch. The language constraint is introduced for the unified experiment. 3) The performance claimed in the original work should be reproducible.

- **Normalized AE** (NAE) (Yoon et al., 2021): AE with an additional mechanism for suppressing the reconstruction of outliers. As in AE, $f(\mathbf{x})$ is the reconstruction error. Only trained with in-distribution data.

Table 1: Evaluation of OOD detectors using the adversarial distributions. Test Set indicates the test split of a test OOD dataset. $D_z$ denotes the dimension of the latent space of the generator. AUC scores are evaluated using 1,000 samples from adversarial distributions.

| OOD Dataset | SVHN | | | | CelebA | | | |
|---|---|---|---|---|---|---|---|---|
| OOD Samples | Test Set | AdvDist | | | Test Set | AdvDist | | |
| $D_z$ | - | 16 | 32 | 64 | - | 16 | 32 | 64 |
| **Weak Detectors** | | | | | | | | |
| Glow | .069 | .018 | .038 | .104 | .542 | .007 | .007 | .087 |
| PXCNN | .076 | .135 | .051 | .119 | .639 | .023 | .088 | .043 |
| AE | .080 | .016 | .055 | .092 | .533 | .050 | .136 | .449 |
| **Strong Detectors** | | | | | | | | |
| NAE | .935 | .227 | .153 | .126 | .874 | .654 | .537 | .470 |
| GOOD | .943 | .399 | .386 | .372 | .939 | .376 | .343 | .300 |
| ACET | .966 | .370 | .398 | .369 | .986 | .382 | .342 | .303 |
| CEDA | .978 | .623 | .628 | .626 | .981 | .618 | .617 | .612 |
| SSD | .989 | .567 | .464 | .453 | .780 | .379 | .410 | .392 |
| MD | .993 | .636 | .619 | .620 | .796 | .417 | .346 | .419 |
| OE | .997 | .700 | .696 | .678 | .992 | .692 | **.697** | **.700** |
| CSI | .998 | **.866** | **.894** | **.986** | .890 | .588 | .582 | .561 |
| **Ensemble** | | | | | | | | |
| GOOD+ACET+CEDA | .983 | .322 | .331 | .316 | .991 | .389 | .397 | .465 |
| MD+NAE+OE | .996 | .723 | .525 | .747 | .978 | **.732** | .684 | .664 |

- **Outlier Exposure** (OE) (Hendrycks et al., 2019a): The predictive confidence of a classifier on an external dataset is minimized.
- **Confidence Enhancing Data Augmentation** (CEDA) (Hein et al., 2019): Similarly to OE, the predictive confidence of a classifier on an external dataset is minimized.
- **Adversarial Confidence Enhancing Training** (ACET) (Hein et al., 2019): The worst-case confidence of a classifier on an external dataset is minimized.
- **Guaranteed OOD Detector** (GOOD) (Bitterwolf et al., 2020): IBP (Gowal et al., 2018) is used to obtain the confidence interval in the neighborhood of OOD data points, and the interval is minimized during training.
- **Mahalnobis Distance** (MD) (Lee et al., 2018): Negative minimum Mahalanobis distance to in-distribution data in the hidden representation spaces is used as $f(\mathbf{x})$. Mahalanobis distances are separately defined for each class and the weight is tuned with a few out-of-distribution data.
- **Self-Supervised Detector** (SSD) (Sehwag et al., 2021): $f(\mathbf{x})$ is computed based on Mahalanobis distance on the representation space which is learned via self-supervised learning.
- **CSI** (Tack et al., 2020): Cosine similarity to the nearest training data in the representation space learned via self-supervised learning is used as $f(\mathbf{x})$.

For OE, CEDA, ACET, and GOOD, the negative maximum softmax probability of a classifier is used as $f(\mathbf{x})$. We utilize the publicly available pretrained models except for MD and SSD where we use the publicly available training scripts. There are a number of other competitive OOD detectors we considered but not included in the experiment of this paper. See Appendix D.2 for discussion on such detectors. We will continuously update our leaderboard afterward to include other OOD detection methods.

## 5.2 IMPLEMENTATION OF ADVERSARIAL DISTRIBUTIONS

**Generator** $g(\mathbf{x})$ We use the decoder of an convolutional AE with the latent space $\mathcal{Z} = [-1, 1]^{D_Z}$ where $D_Z = \{16, 32, 64\}$. The autoencoder is trained to reconstruct samples in $\mathcal{D}_{out}$.

**Binary Classifier** $h(\mathbf{x})$ Our binary classifier $h(\mathbf{x})$ is based on ResNet-50 pre-trained on ImageNet. The last layer is replaced with a newly initialized binary classification layer, and then the whole

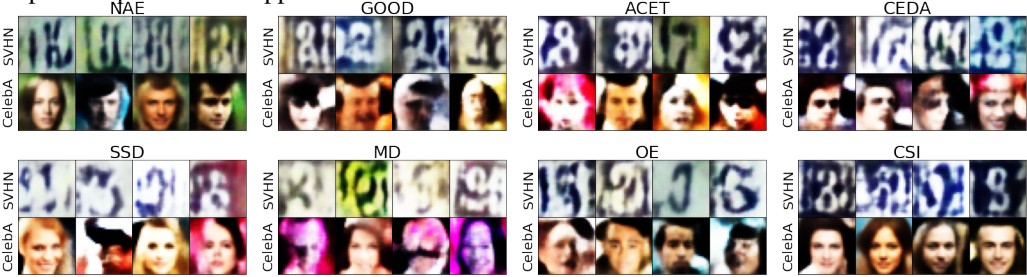

Figure 3: Random samples from the adversarial distribution of the weak detectors ($D_Z = 32$). More samples are provided in Appendix F.

Figure 4: Random samples from adversarial distributions of the strong detectors. More samples can be found in Appendix F.

network is fine-tuned to classify $\mathcal{D}_{in}$ from $\mathcal{D}_{out}$. We use the logit value as $h(\mathbf{x})$. As a zero-inlier threshold, we set $\eta_h = -5$ for SVHN and $\eta_h = -1$ for CelebA.

**Adversarial Distribution** $p_f(\mathbf{x})$  We apply standardization on $f(\mathbf{x})$. The mean and the standard deviation of detector score is computed on the validation split of in-distribution data. For all experiments, the temperature is set $T = 0.1$.

Gibbs sampling is performed on $\mathcal{Z}$ to generate samples from $p_f(\mathbf{x})$. At each step, a dimension of $\mathbf{z}$ is randomly selected, and a proposal is generated by a unit Gaussian distribution. A proposal outside the domain is projected back. A proposal is accepted based on Metropolis-Hastings criterion. We run 1,000 independent Markov chains with each chain runs for 10,000 steps. The final states of Markov chains are accepted as samples. Samples that do not belong to $\mathcal{T}_h$ are rejected. More implementation details can be found in Appendix C.

## 5.3 ADVERSARIAL DISTRIBUTIONS AGAINST DETECTORS WITH KNOWN WEAKNESSES

We confirm the effectiveness of adversarial distributions by applying them on the weak detectors, Glow, PXCNN, and AE, where the weaknesses have already been analyzed. The likelihoods of Glow and PXCNN become spuriously high for images with low complexity (Serrà et al., 2020). Similarly, AE produces also low reconstruction error simple images (Yoon et al., 2021). From Figure 3, the samples from adversarial distributions $p_f(\mathbf{x})$ are consistent with the previously known tendency. The samples are very blurry and a large portion of pixels are monotone. Simplicity of the samples are more clear when they are compared to the samples from a strong detector's $p_f(\mathbf{x})$ in Figure 4.

## 5.4 EVALUATING STATE-OF-THE-ART DETECTORS

The proposed methods reveals that 8 strong detectors, with almost equivalent and near-perfect performance on CIFAR-10 vs SVHN, are actually very vulnerable outside of the test points. All the strong detectors show significantly decreased AUC against samples from adversarial distributions. Among the tested detectors, CSI is the most robust when SVHN is $\mathcal{D}_{out}$. However, CSI does not perform well when outliers and adversarial distributions are based on CelebA. All other detectors show a similar degree of performance drop against samples from their adversarial distributions $p_f(\mathbf{x})$ visualized in Figure 4. Note that most of samples from $p_f(\mathbf{x})$ retain the visual features of the OOD dataset on which $p_f(\mathbf{x})$ based. Also, the samples are visually dissimilar to CIFAR-10.

In the range from 16 to 64, the choice of latent space dimensionality $D_Z$ does not significantly affect the efficacy of adversarial distributions in terms of AUC scores. Using smaller $D_Z$ makes the search space narrower, but MCMC can be performed more effectively. Meanwhile, larger $D_Z$ increases the diversity of samples, but MCMC becomes challenging.

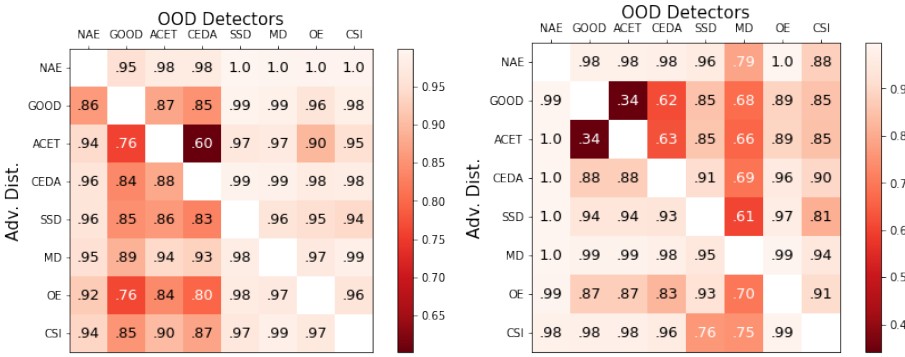

Figure 5: OOD detection AUC scores evaluated against other detectors' adversarial distribution samples. (Left) SVHN $D_z = 32$ (Right) CelebA $D_z = 32$

**Transferability and Detector Ensemble** We question whether an adversarial distribution is transferable, i.e., given two detectors $f_1(\mathbf{x})$ and $f_2(\mathbf{x})$, are samples from $p_{f_1}(\mathbf{x})$ able to deceive another detector $f_2(\mathbf{x})$? We collect samples from $p_{f_1}(\mathbf{x})$ then check if $f_2(\mathbf{x})$ can detect them as outliers. The pairwise results are displayed in Figure 5, showing that there are varying degree of transferability. Adversarial distributions are largely transferable between GOOD, CEDA, and ACET, possibly because their constructions are significantly similar (Hein et al., 2019). Except for the three, the other detectors are generally capable of detecting another detector's adversarial distribution samples. It is interesting to see that NAE also performs well on detecting adversarial distribution samples from other detectors, while NAE is not the best performing detector. We suspect the limited transferability is originated from the distinct inductive biases of detectors. For example, NAE is the only autoencoder-based algorithm, and SSD and CSI are trained via different self-supervised learning processes.

When detectors do not share failure modes, a more robust detector may be formed through averaging their (normalized) detection scores. We construct two ensembles of three OOD detectors, one with highly correlated failure modes (GOOD+ACET+CEDA) and the other with less correlated failure modes (MD+NAE+OE), and measure their robustness. From the last two rows in Table 1, both ensembles barely improve the robustness of OOD detector compared to the individual detectors.

# 6 DISCUSSION

**Extensions** The outlier space utilized in the analysis can be expanded by augmenting the test OOD dataset with an additional OOD dataset. Moreover, there is significant flexibility in the design of $g(\mathbf{x})$. In a specific application, domain knowledge can be embedded in $g(\mathbf{x})$ to provide a more meaningful outlier space.

**Limitations** Our methods still rely on test OOD datasets to obtain $g(\mathbf{x})$ and $h(\mathbf{x})$, and the choice of the datasets need human design. Furthermore, operating in the reduced outlier space, our methods can not confirm that a detector is optimal. However, these limitations are also present in the current evaluation protocol as well.

**Future Work** To facilitate the development of OOD detection methods, we intend to publish an online software suite for evaluating OOD detectors based on adversarial distributions. We also plan to publish an online leaderboard displaying the results from an adversarial distribution analysis, and to update the leaderboard with the latest OOD detection methods.

# 7 CONCLUSION

In this paper, we have addressed the limitations of the current evaluation protocol for OOD detection and proposed a novel framework, adversarial distributions, that can be used to investigate failure modes of OOD detectors. The proposed framework posing new challenges in OOD detection by discovering unexplored weaknesses in the existing OOD detectors, We expect our framework to stimulate the advance of trustworthy machine learning, where a model needs to be evaluated beyond the fixed test dataset.

ETHICS STATEMENT

Our main ethical concern is that a subset of OOD detectors used in our experiment, OE, CEDA, ACET and GOOD, are trained using 80 Million Tiny Images dataset (Torralba et al., 2008), which is retracted by authors over ethical concerns. While we were aware of the issue of the dataset, the use of models trained on the dataset was inevitable because of the reproducibility. We have tried to train OOD detectors using alternative datasets but failed to reach the performance of detectors originally trained on 80 Million Tiny Images. To minimize the effect of the retracted dataset, the dataset was never used directly. We only used the publicly available model checkpoints, and did not download or access to a copy of dataset.

REPRODUCIBILITY STATEMENT

As mentioned in intro, we will provide the software suite to evaluate the weaknesses of the OOD detector. Since our algorithm consists of widely used techniques such as autoencoders, image classifiers, and MCMC sampling, there will be no problem with reproduction.

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

APPENDIX

Appendix is organized as follows:

- Section A: Proof of Proposition 1
- Section B: Datasets
- Section C: Implementation of Adversarial Distributions
- Section D: Implementation of OOD Detectors
- Section E: Extended Experimental Results
- Section F: Additional Samples from Adversarial Distributions

## A   PROOF OF PROPOSITION 1

**Proposition 1** (Restated)**.** Suppose two probability distributions $P$ and $Q$ with their supports $\mathcal{S}_P$ and $\mathcal{S}_Q$, respectively. $P$ and $Q$ are OOD to each other, if and only if there exists a binary classifier $h(\mathbf{x}) : \mathcal{X} \to \mathbb{R}$ that perfectly discriminates $P$ and $Q$, i.e., there exists a threshold $\eta_h \in \mathbb{R}$ such that the classification boundary $h(\mathbf{x}) = \eta_h$ separates $\mathcal{S}_P$ and $\mathcal{S}_Q$.

*Proof.* If $P$ and $Q$ are OOD to each other, then $\mathcal{S}_P \cap \mathcal{S}_Q = \varnothing$. Then, we can construct a classifier $h(\mathbf{x})$ such that $h(\mathbf{x}) > \eta_h$ for $\mathbf{x} \in \mathcal{S}_P$ and $h(\mathbf{x}) \leq \eta_h$ for $\mathbf{x} \in \mathcal{S}_Q$ for any constant $\eta_h \in \mathbb{R}$. Therefore, $h(\mathbf{x}) = \eta_h$ separates $\mathcal{S}_P$ and $\mathcal{S}_Q$.

Given a classifier $h(\mathbf{x})$ such that $h(\mathbf{x}) > \eta_h$ for $\mathbf{x} \in \mathcal{S}_P$, suppose that $P$ and $Q$ are not OOD to each other. Then, there exists $\mathbf{x}^*$ such that $\mathbf{x}^* \in \mathcal{S}_P$ and $\mathbf{x}^* \in \mathcal{S}_Q$ at the same time. However, $h(\mathbf{x}^*)$ can not be $h(\mathbf{x}) > \eta_h$ and $h(\mathbf{x}) \leq \eta_h$ at the same time. Due to the contradiction, given a perfectly classifier, $P$ and $Q$ should be OOD to each other. □

## B   DATASETS

In our experiments, we use CIFAR-10, SVHN, and CelebA to demonstrate the proposed methods and to evaluate OOD detection algorithms, and their details are described in Table 2. In all cases, we use the official test split as our test datasets. For CIFAR-10 and SVHN, the official train split is randomly splitted into train (90%) and valid split (10%). For CelebA, we use the official train-valid split as it is given. Each CelebA image is center-cropped into a $140 \times 140$ image and scaled to $32 \times 32$ using the bilinear interpolation.

Table 2: Statistics for datasets.

| Dataset | Train | Valid | Test |
|---|---|---|---|
| CIFAR-10 (Krizhevsky, 2009) | 45,000 | 5,000 | 10,000 |
| SVHN (Netzer et al., 2011) | 65,930 | 7,327 | 26,032 |
| CelebA (Liu et al., 2015) | 162,770 | 19,867 | 19,962 |

No augmentation is applied to test data. However, random horizontal flip with probability 50% and uniform dequantization (Theis et al., 2015) are applied to training images in the training of the binary classifiers and the autoencoders in the adversarial search and adversarial distributions.

## C   IMPLEMENTATION OF ADVERSARIAL DISTRIBUTIONS

### C.1   IMPLEMENTATION OF BINARY CLASSIFIERS AND AUTOENCODERS

**Binary Classifiers**   We train binary classifiers $h(\mathbf{x})$ that classify $\mathcal{D}_{in}$ from $\mathcal{D}_{out}$. Two binary classifier are trained with different $\mathcal{D}_{out}$'s. For one classifier, $\mathcal{D}_{out}$ is SVHN, and for the other, $\mathcal{D}_{out}$ is CelebA. $\mathcal{D}_{in}$ is CIFAR-10 for both classifiers. Both classifiers have an identical architecture. We replace the last layer of ResNet-50 pre-trained on ImageNet with a newly initialized binary classification layer and fine-tune the whole network using binary cross entropy loss. The pre-trained model is provided by TorchVision. Training is performed for 100 epochs using Adam with the learning rate

of $1 \times 10^{-5}$, where an epoch is defined with respect to the smaller of $\mathcal{D}_{in}$ or $\mathcal{D}_{out}$. Each mini-batch consists of 128 samples from each of $\mathcal{D}_{in}$ and $\mathcal{D}_{out}$. During training, validation loss is recorded and model weights with the best validation loss are used in our experiments.

**Autoencoders** The decoder of a convolutional autoencoder is used as the generator $g(\mathbf{z})$. The architecture of the autoencoder is similar to what is used in (Ghosh et al., 2020) and is described in Table 3. The latent space of this autoencoder is given as $\mathcal{Z} = [-1, 1]^{D_z}$ where $D_z = \{16, 32, 64\}$.

Table 3: Autoencoder architecture. $\text{Conv}_N(\text{M})$ indicates a 2D convolution operation with a $N \times N$ kernel and $M$ output channels. BN denotes batch normalization, and ReLU means the rectified linear unit activation.

| Encoder | Decoder |
| --- | --- |
| $\text{Conv}_4(128)$-BN-ReLU- | $\text{ConvT}_8(1024)$-ReLU- |
| $\text{Conv}_4(256)$-BN-ReLU- | $\text{ConvT}_4(512)$-ReLU- |
| $\text{Conv}_4(512)$-BN-ReLU- | $\text{ConvT}_4(256)$-ReLU- |
| $\text{Conv}_4(1024)$-BN-ReLU- | $\text{ConvT}_1(3)$-Sigmoid |
| FC(1024)-BN-ReLU- | |
| FC(32)-Tanh | |

We train the autoencoder with samples in the union of $\mathcal{D}_{in} \cup \mathcal{D}_{out}$, where $\mathcal{D}_{out}$ is SVHN or CelebA. Autoencoders are trained for 500 epochs using Adam with the learning rate of $1 \times 10^{-4}$.

## C.2 IMPLEMENTATION OF ADVERSARIAL DISTRIBUTIONS

Here, we describe the method for sampling from the practical version of adversarial distribution (Eq. (2)).

**Enforcing Support Constraint** In Eq. (2), the constraint $\mathbf{z} \in \mathcal{T}_h$ needs to be enforced, while most MCMC algorithms for real-valued states assume an unbounded space. We enforce the constraint by two-staged approach.

First, instead of the original adversarial distribution in Eq. (2), we run MCMC for an alternative distribution $p'_f(\mathbf{z}; g, h)$, where two barrier functions, $H_1(\mathbf{z})$ and $H_2(\mathbf{z})$, are added to the energy function.

$$p'_f(\mathbf{z}; g, h) = \frac{1}{Z'(f, g, h)} \exp(-E(\mathbf{z})/T), \quad \mathbf{z} \in \mathbb{R}^{D_z}, \tag{6}$$

$$E(\mathbf{z}) = f(g(\mathbf{z})) + H_1(\mathbf{z}) + H_2(\mathbf{z}), \tag{7}$$

$$H_1(\mathbf{z}) = \max(0, h(f(\mathbf{z})) - \eta_h), \tag{8}$$

$$H_2(\mathbf{z}) = 10 \sum_{d=1}^{D_z} \max(0, |\mathbf{z}_{(d)}| - 1)^2, \tag{9}$$

where $\mathbf{z}_{(d)}$ is $d$-th element of a vector $\mathbf{z}$, and $Z'(f, g, h) = \int \exp(-E(\mathbf{z})/T)d\mathbf{z}$. $H_1(\mathbf{z})$ drives a Markov chain into $\{\mathbf{z}|f(\mathbf{z}) \leq \eta_h\}$ by assigning higher energy on $\mathbf{z}$ that violates the constraint. Similarly, $H_2(\mathbf{z})$ assigns higher energy for $\mathbf{z}$ outside $[-1, 1]^{D_z}$. Due to $H_1(\mathbf{z})$ and $H_2(\mathbf{z})$, samples that comply the constraint $\mathbf{z} \in \mathcal{T}_h$ appear more frequently in MCMC. Note that $H_1(\mathbf{z})$ and $H_2(\mathbf{z})$ have non-zero only when the constraint is violated, and the $E(\mathbf{z}) = f(g(\mathbf{z}))$ for $\mathbf{z} \in \mathcal{T}_h$.

Second, after running MCMC, we reject any sample that does not obey the constraint $\mathbf{z} \in \mathcal{T}_h$. This rejection process ensures that the generated samples are from the original adversarial distribution Eq. (2).

**Metropolis-Hasting Acceptance** To generate samples from an adversarial distribution (Eq. (2)), we apply Metropolis-Hastings algorithm (Metropolis et al., 1953) in $\mathcal{Z}$. An initial state $\mathbf{z}_0$ is drawn from an uniform distribution. Given a state $\mathbf{z}_t$ at time $t$, a candidate for the next state $\bar{\mathbf{z}}_{t+1}$ is generated by the proposal distribution. We use a Gaussian distribution which is centered at $\mathbf{z}_t$ and

has the fixed standard deviation of 0.1 as our proposal distribution. A proposed sample is accepted with the probability of $\min\{1, \exp((E(\bar{\mathbf{z}}_{t+1}) - E(\mathbf{z}_t))/T)\}$, yielding $\mathbf{z}_{t+1} = \bar{\mathbf{z}}_{t+1}$. Otherwise, $\mathbf{z}_{t+1} = \mathbf{z}_t$. We take the final state of a Markov chain as a generated sample. As stated in the previous paragraph, the final state is rejected if $\mathbf{z} \notin \mathcal{T}_h$. An accepted sample is mapped to $\mathcal{X}$ by $\mathbf{x} = f(\mathbf{z})$.

## C.3 Ensemble

An ensemble OOD detector is built by combining predictions from multiple OOD detectors. First, each OOD detector's detector score is standardized separately. For standardization, the mean and the standard deviation of each detector score is computed on the validation in-distribution dataset. The standardization process is required to account the scale difference between OOD detectors. Then, the standardized detector score is averaged to yield the detector score for the ensemble detector.

## D Implementation of OOD Detectors

### D.1 Implementation of Weak and Strong Detectors

In this section, we describe how OOD detectors are implemented in our experiments.

**AE** The overall architecture of AE is identical to what is described in Section C but with a different dimensionality of the latent space $D_Z = 128$. AE is trained for 500 epochs on in-distribution training set using Adam optimizer with the learning rate $1 \times 10^{-4}$. A mini-batch contains 128 samples. A model with the smallest reconstruction error on validation split is selected.

**PXCNN** PXCNN is implemented based on an open-sourced code repository[1] and re-trained on CIFAR-10. All the model parameters are set to the default values, i.e., `nr_resnet=5`, `nr_filters=80`, `nr_logistic_mix=10`, `resnet_nonlinearity='concat_elu'`. An input image is linearly scaled into $[-1, 1]$, and horizontal flipped with the probability of 50%. The model is trained for 200 epochs where a mini-batch contains 64 samples. We use Adam optimizer with the learning rate $1 \times 10^{-4}$ which is decayed by a multiplicative factor of 0.999995 every iteration.

**Glow** Glow is implemented based on two repositories[2] [3] and traianed on CIFAR-10. Our Glow model utilizes a multi-scale architecture with three levels of the latent representations are present, i.e., `L=3`. Each level contains 32 flow steps. We use $1 \times 1$ invertible convolution and ActNorm with the scale of 1.0. The model is trained for 500 epochs with the batch size of 64. Adam optimizer with the learning rate $1 \times 10^{-4}$ is utilized. The gradient is clipped so that its norm is no larger than 0.1.

**NAE** The pre-trained model is downloaded from the official repository[4]. We use Conv32Big version, which showed better performance on CIFAR-10 vs SVHN.

**OE** Among the available pre-trained models in the official repository[5], we select the best performing version, i.e., `cifar10_allconv_oe_scratch_epoch_99.pt`. We normalize an input image with `means=(0.4914, 0.4822, 0.4465)` and `std=(0.2471, 0.2435, 0.2615)`, where each component corresponds to RGB, respectively.

**CEDA, ACET, GOOD** The pre-trained models are provided by the official repository of GOOD[6]. Among multiple versions of GOOD, we use GOOD80, as recommended in the original paper.

---

[1] `https://github.com/pclucas14/pixel-cnn-pp`
[2] `https://github.com/chaiyujin/glow-pytorch`
[3] `https://github.com/chrischute/glow`
[4] `https://github.com/swyoon/normalized-autoencoders`
[5] `https://github.com/hendrycks/outlier-exposure`
[6] `https://gitlab.com/Bitterwolf/GOOD`

**MD**  We implement MD following the official repository[7]. Based on the pre-trained ResNet-32 provided by the official repository, we train the weights of each layer where Mahalanobis distance is computed. 1,000 SVHN images are used to determine such weights. Note that using test OOD dataset during the training is usually not acceptable in a typical OOD detection setting.

**SSD**  We use the official training script to train SSD [8].

**CSI**  We download the unlabelled multi-class CIFAR-10 model from the official repository[9].

### D.2  DETECTORS NOT INCLUDED IN EXPERIMENTS

We have considered other OOD detectors such as Likelihood Ratio (Ren et al., 2019), Input Complexity (Serrà et al., 2020), Deterministic Uncertainty Quantification (DUQ) (Van Amersfoort et al., 2020), and Likelihood Regret (Xiao et al., 2020). However, we could not include them in our experiments for the following reasons.

**Likelihood Ratio**  While there is the official public repository written in TensorFlow[10], we have failed to reproduce the result in PyTorch.

**Input Complexity**  We confirm that the log-density estimates from generative models are correlated to the bits after compression, but fail to achieve AUC score higher than 0.9 in CIFAR-10 vs SVHN setting. For generative models, PXCNN and Glow are tested, and for image compression algorithms, PNG, JPEG2000, and FLIF are tried. Our generative models are implemented as Section C. For PNG and JPEG2000, we use OpenCV[11], and for FLIF, we use imageio-flif[12]. No combination of a generative model and a compression algorithm results in AUC higher than 0.9.

**DUQ**  Investigating the official repository [13], we find that different normalization transforms are applied to in-distribution (CIFAR-10) and test OOD dataset (SVHN). CIFAR-10 is normalized with parameters `mean=(0.4914, 0.4822, 0.4465)` and `std=(0.2023, 0.1994, 0.2010)`, whereas SVHN is normalized using `mean=(0.5, 0.5, 0.5)` and `std=(0.5, 0.5, 0.5)`. Applying different transformations makes classification between CIFAR-10 and SVHN easier. When we modify the code so that an identical transformation is applied to both datasets, AUC score drops below 0.9. Therefore, we do not include DUQ in our experiments.

**Likelihood Regret**  The method is very slow, particularly because it does not support batch processing, i.e., only one sample can be processed at a time. We decide that the method is too slow to apply the adversarial distribution approach.

---

[7]https://github.com/pokaxpoka/deep_Mahalanobis_detector
[8]https://github.com/inspire-group/SSD
[9]https://github.com/alinlab/CSI
[10]https://github.com/google-research/google-research/tree/master/genomics_ood/images_ood
[11]https://opencv.org/
[12]https://codeberg.org/monilophyta/imageio-flif
[13]https://github.com/y0ast/deterministic-uncertainty-quantification

## E    EXTENDED EXPERIMENTAL RESULTS

### E.1    BINARY CLASSIFICATION

Here, we show the effect of test-time augmentation and report the zero-inlier threshold. Test-time augmentation dramatically increases the zero-inlier threshold, meaning that the classifiers become more robust.

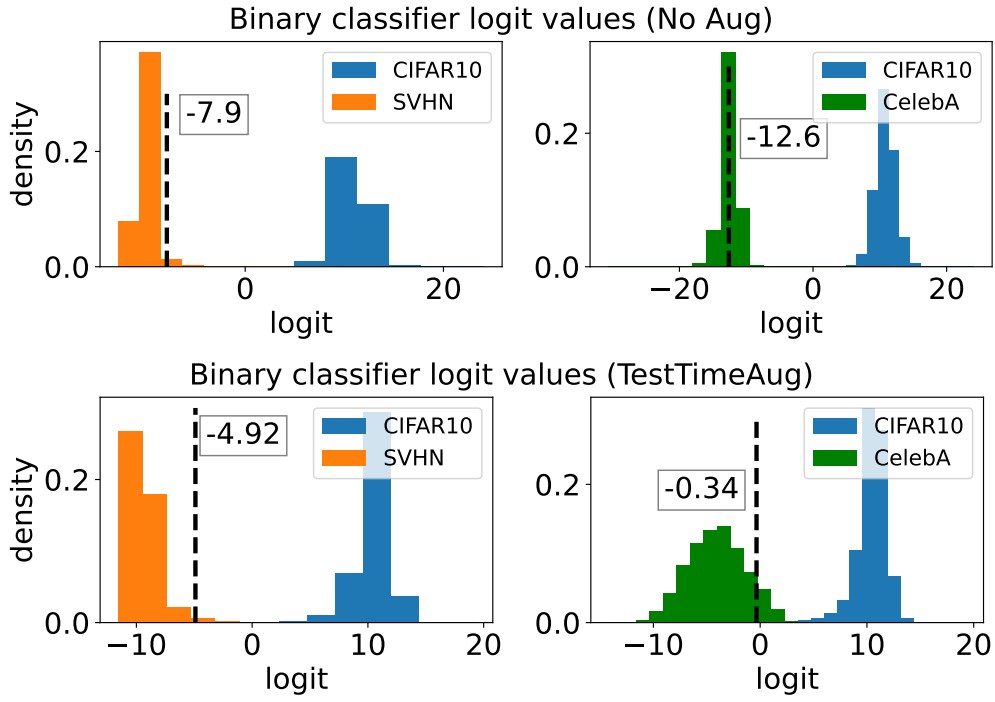

Figure 6: Distribution of the logit values in binary classification. The zero-inlier threshold, the smallest logit of all inliers are denoted as dotted lines.

### E.2    RUNTIME AND COMPUTING ENVIRONMENT

Table 4 shows time required to perform MCMC sampling and for each OOD detectors. For MCMC, time required for running 10,000 steps of MH update is shown. The main factor that determines the runtime is the inference time of an OOD detector. Inference of all OOD detector is performed on a single Tesla V100 GPU.

Table 4: Time required for performing a single run of MCMC

| Detector | MCMC |
|---|---|
| Glow | 3hr 15min |
| PXCNN | 10min |
| AE | 1hr 10min |
| NAE | 50min |
| GOOD | 22min |
| ACET | 22min |
| CEDA | 22min |
| SSD | 37min |
| MD | 1hr 30min |
| OE | 22min |
| CSI | 20hr |

## F ADDITIONAL SAMPLES FROM ADVERSARIAL DISTRIBUTIONS

From Figure 7 to Figure 12 present samples from adversarial distributions of each OOD detectors.

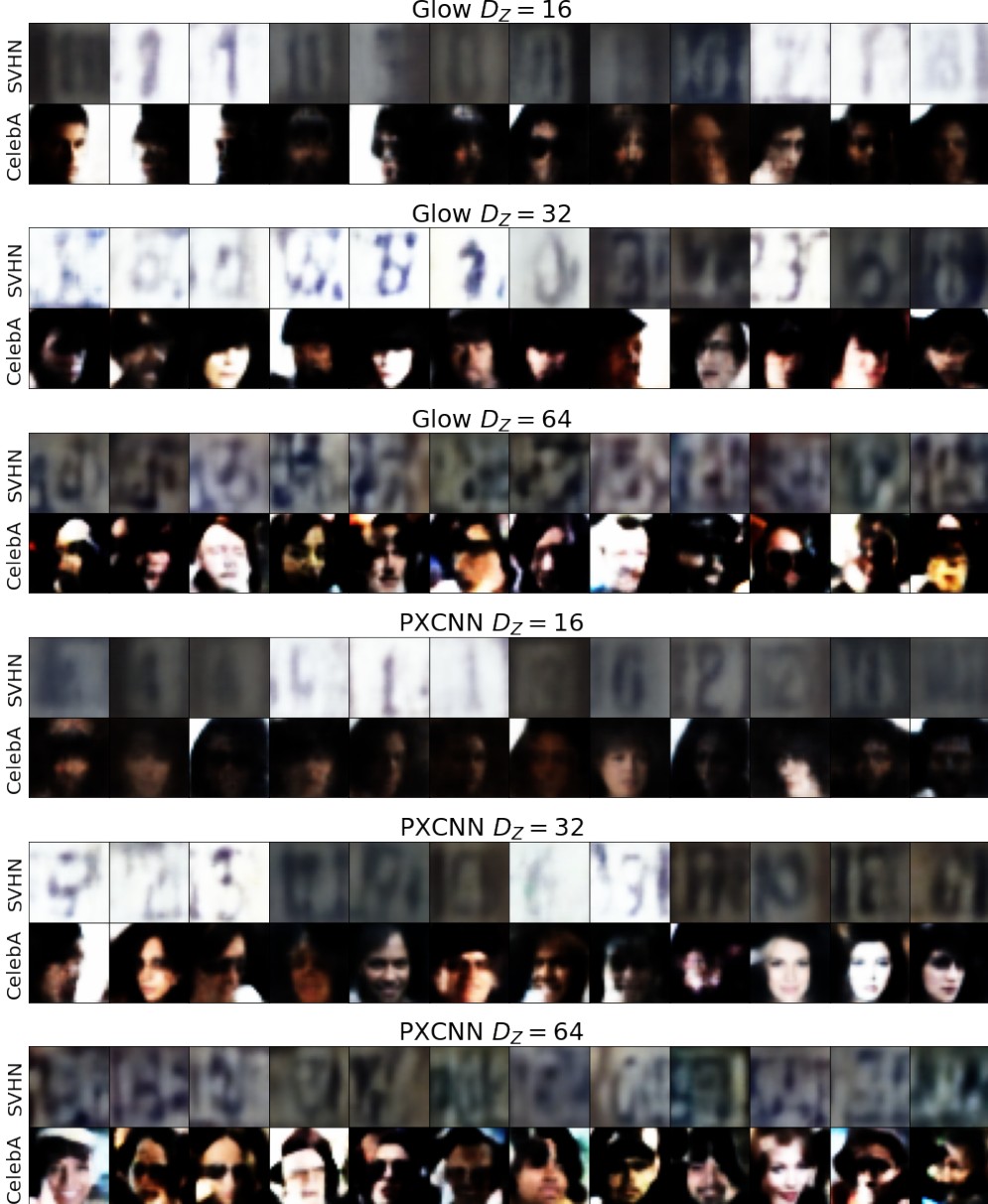

Figure 7: Samples from adversarial distributions.(1/6)

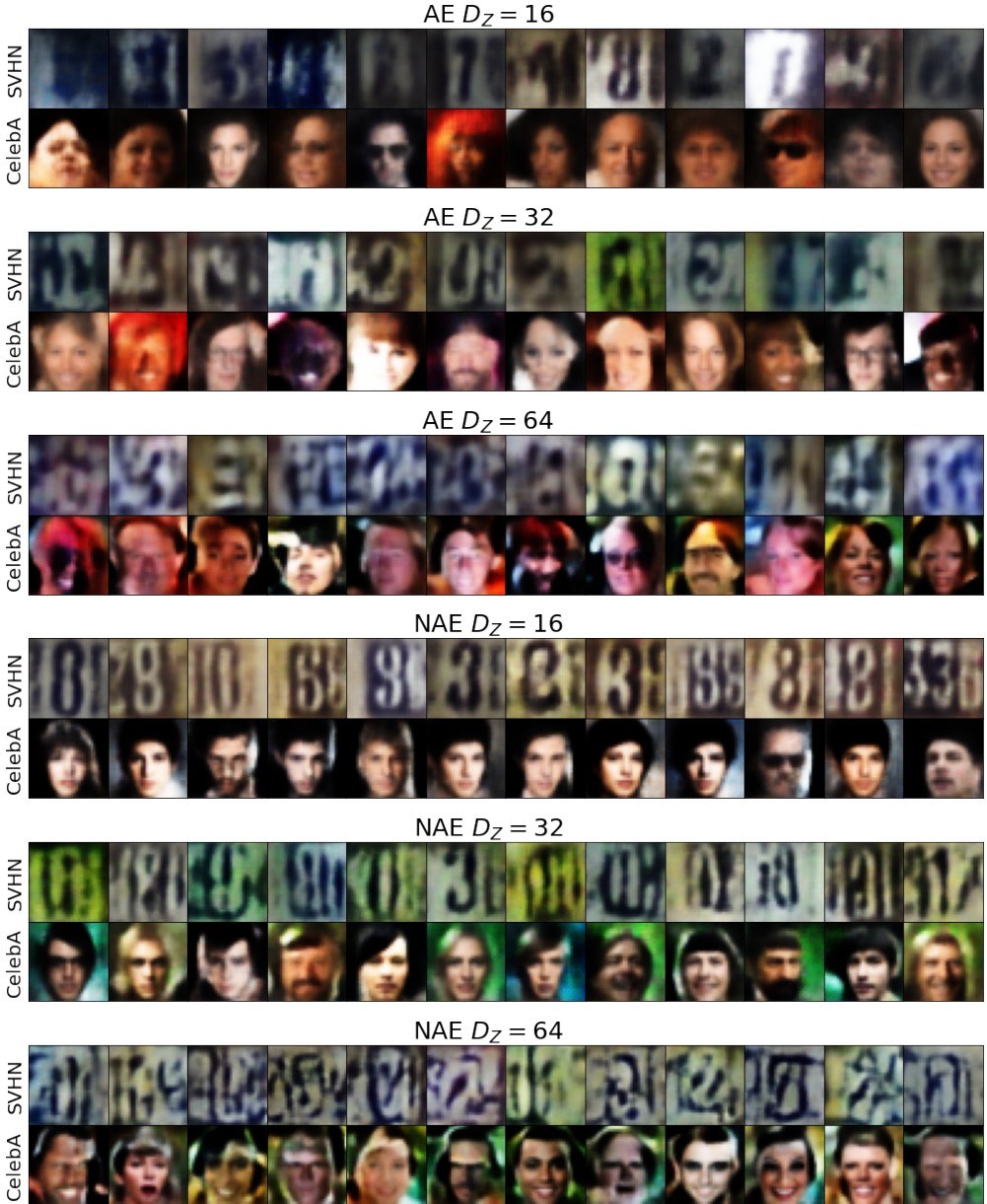

Figure 8: Samples from adversarial distributions.(2/6)

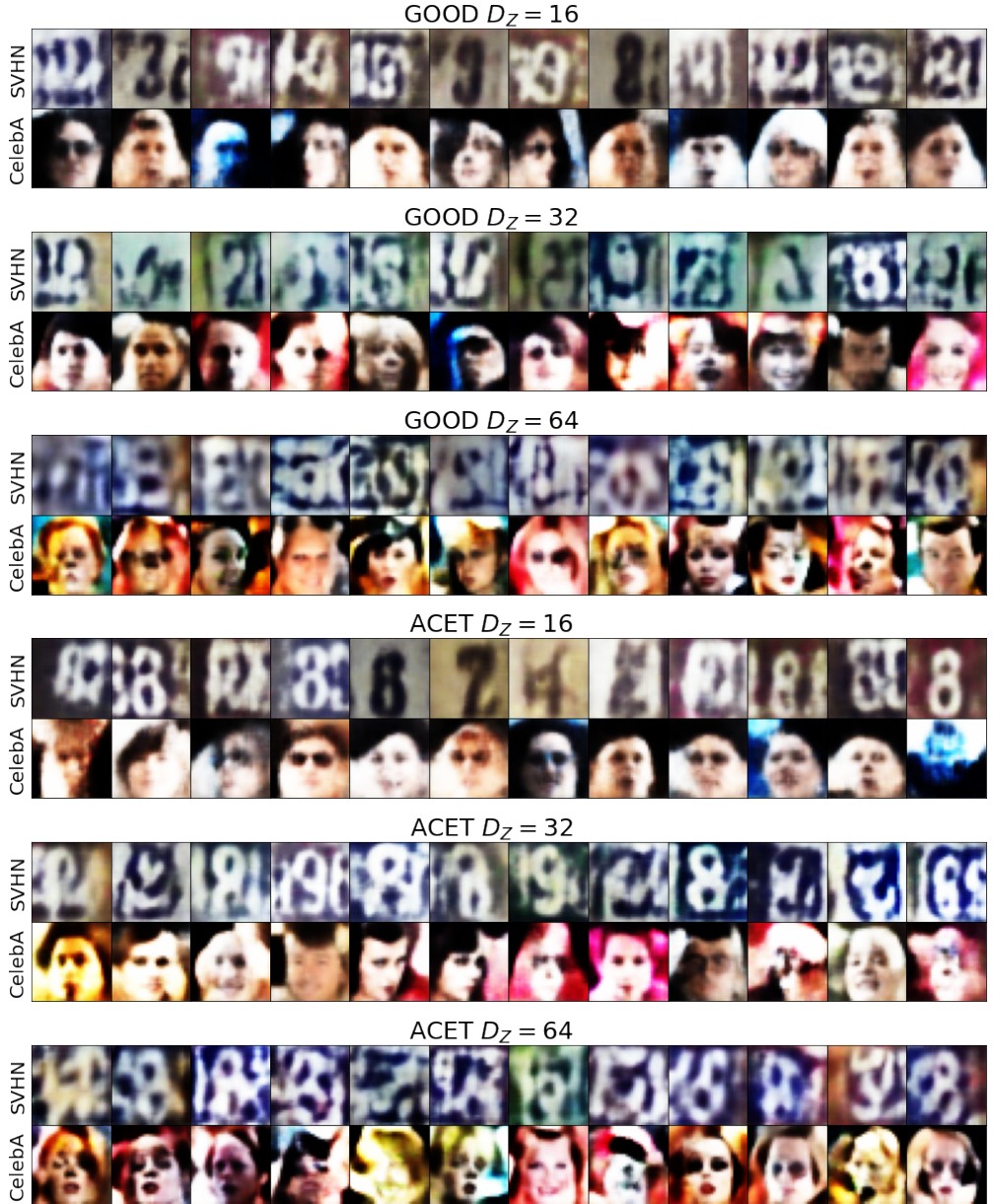

Figure 9: Samples from adversarial distributions.(3/6)

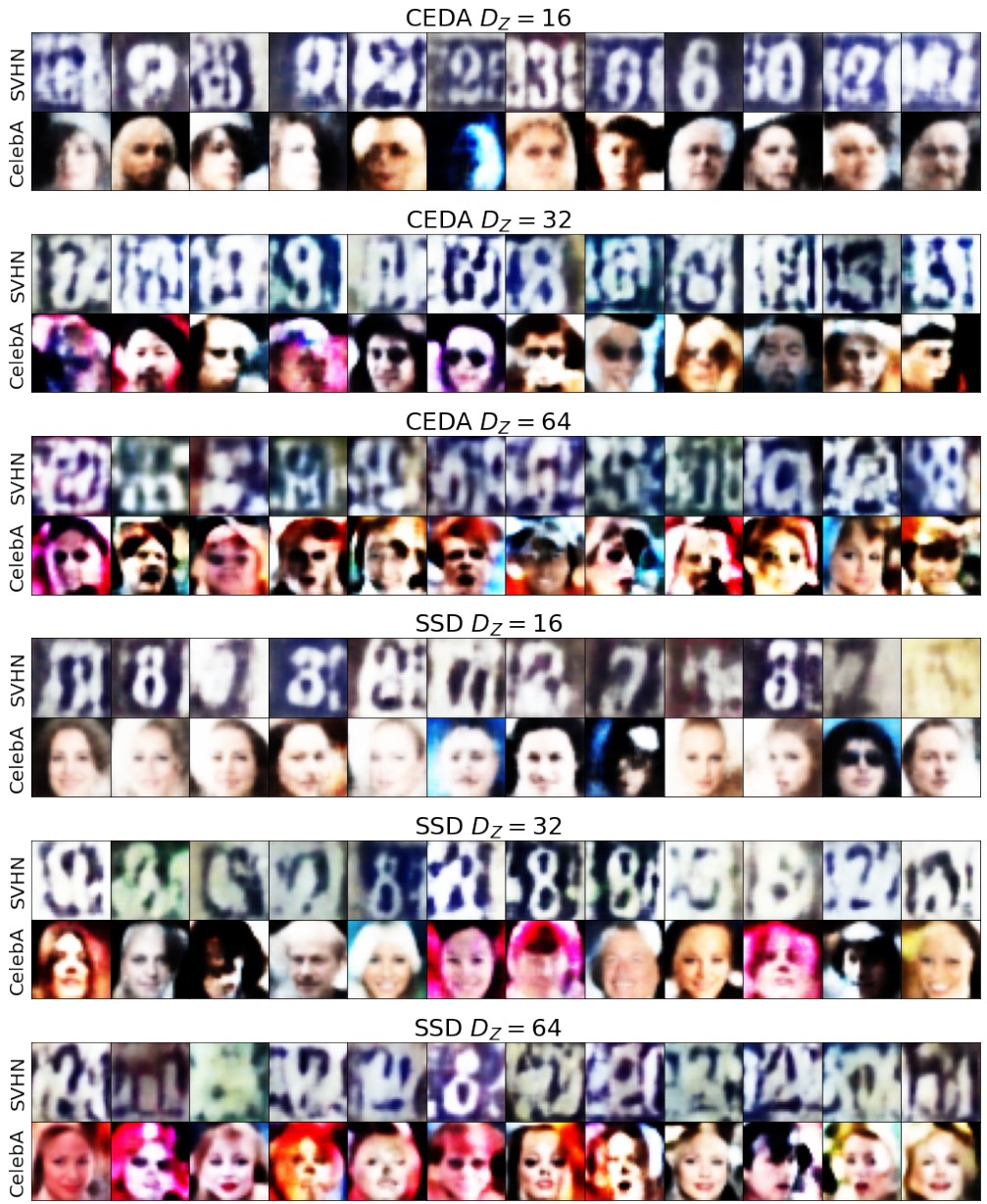

Figure 10: Samples from adversarial distributions.(4/6)

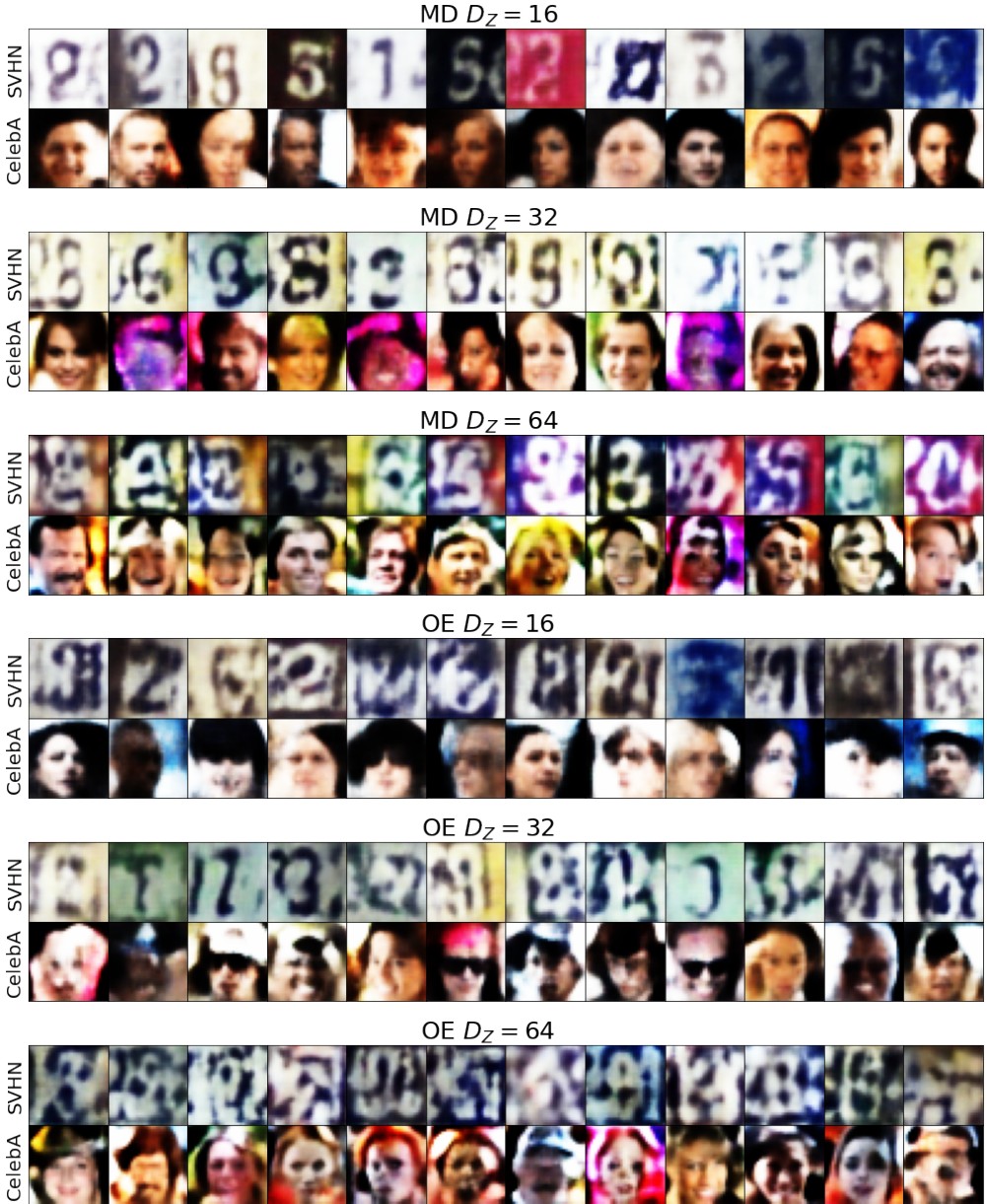

Figure 11: Samples from adversarial distributions.(5/6)

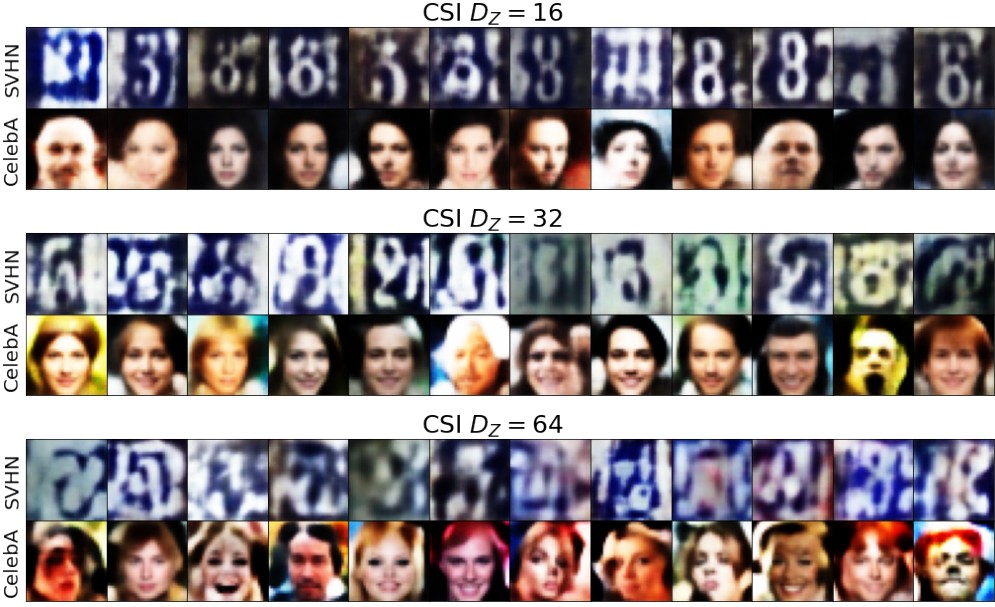

Figure 12: Samples from adversarial distributions.(6/6)

