# OpenReview forum: "Adversarial Distributions Against Out-of-Distribution Detectors"
_ICLR.cc/2022/Conference — ICLR 2022 Submitted_

### Official Review · Reviewer_XMWK · 2021-10-30

**Correctness:** 2
**Technical Novelty And Significance:** 2
**Empirical Novelty And Significance:** 2
**Recommendation:** 3
**Confidence:** 4

**Main Review:**

This paper contains some interesting ideas, and in general I like the fact the derivations and experiments are clearly presented.

However, a major problem I feel is that it has an unclear relationship with previous work. This is far from the first work that consider constructing a distribution that is "adversarial" to the OOD. For example, Hein's group has produced a line of work in this direction, and also recently there have been many other works (e.g., https://arxiv.org/abs/2006.15207). For a big part of the paper, these related works are not discussed at all, and in fact Bitterwolf's work is discussed only until page 7. This does not seem to be the right way of handling related work.

I guess this work may be arguing that, even though the previous works have considered "adversarially constructed distributions against OOD", this work is more explicit and systematic about this direction. To that end I remain a bit skeptical. It is not the case that, just by considering larger and larger space, it "automatically" makes an attacking-OOD paper more interesting. In fact, a small space of perturbation that leads to failure of OOD is even more interesting, than a very large space (clearly, NN will likely be poorly performing on far-away inputs, given the state-of-the-art technology).

I am also worried about the very strong claims made in the paper, such as "In this paper, we have addressed the limitations of the current evaluation protocol for OOD detection and proposed a novel framework, adversarial distributions, that can be used to investigate failure
modes of OOD detectors." Do we really have addressed the problem (so we cannot imagine any further work to improve the evaluation, and so this work has exhausted the entire space?)

Nit on the naming. Adversarial search induces an adversarial distribution from the original "natural distribution", so I don't really think it is a good name to call "adversarial search" vs. "adversarial distribution" in the definitions. I don't really have a good suggestion, but how about "adversarial sampling"?

**Summary Of The Paper:**

This paper studies the problem of constructing adversarial distributions to fool "naturally trained" OOD detectors. A optimization-based and a sampling-based method are proposed and evaluated respectively.

**Summary Of The Review:**

I am worried about the handling of previous work in this paper, and it seems to me to be overselling its contributions. For these reasons, I would vote for rejection.

---

> ### Author Response · Authors · 2021-11-23
> **Response for Reviewer XMWK**
>
>
> Dear Reviewer XMWK,
>
> We thank you for your detailed comment.
>
> **1. Literature Review**
> In retrospect, we have to agree with the reviewer's comment that there is much room for improvement in our literature review. We intend to update the manuscript with a more extensive
> list of related previous works, beginning with obviously missed references such as ATOM (Chen et al., 2021). Also, to be able to better place our contributions in the context of the literature, particularly with respect to the many works from Hein’s group, we have moved the related work section toward the end of the introduction. In the original manuscript we had discussed and benchmarked our experiments against the many works from Hein's group including CEDA, ACET,
> and GOOD, and also discussed Bitterwolf's work in some depth in page 4 (Section 3.2) and page 7, but the reviewer is correct that we did not properly discuss and acknowledge these works in the introduction.
>
>
> **2. Exploring Larger Outlier Spaces**
> We also agree that a small perturbation making a detector fail is indeed intriguing. Meanwhile, larger outlier spaces beyond small balls are possibly equally interesting, because the space has never been explored before, and the exploration of the space may require different tools.
> The main challenge of exploring the space beyond norm-balls is that there has been no mechanism that can ensure a larger space contains only outliers.
> In our paper, we enable the exploration by providing multiple measures to assure that the space contains no inliers.
> As a result, we could find multiple examples that are obviously OOD to human eyes but can fool OOD detectors.
>
> Furthermore, from the adversarial attack perspective, a threat model, i.e., the set of possible attacks, does not need to be confined in a small ball. There are a number of adversarial attack algorithms, including adversarial patch (Brown et al., 2017) and adversarial eye glasses (Sharif et al., 2018), that implement threat models different from norm balls.
> However, there is little previous work that investigates threat models beyond norm balls in the OOD detection setting.
> As an ideally robust OOD detector should be robust against any threat model, we believe that exploring attacks beyond the ball can be a meaningful contribution to the community.
>
> As an additional comment, neural networks that fail by producing overconfidence predictions on far-away inputs are ReLU networks (Hein et al., 2019). It is true that some of the OOD detectors tested in our experiments, e.g., OE, CEDA, and ACET, are ReLU networks. However, detector scores of other OOD detectors are not computed through the softmax output of ReLU networks. For example, the MD detector utilizes the Mahalanobis distance which naturally assigns low confidence (i.e., high detector score) on far-away inputs.
>
>
>
> **3. Addressing OOD Detection Evaluation**
> Please note that the "Limitations" paragraph in Section 6 states that our method only operates in a subset of the whole outlier space and hence is not capable of determining whether an OOD detector is optimal.
> This limitation could have been stated more explicitly throughout the paper, and we would like to update the writing to prevent possible misunderstanding regarding what the proposed method is capable of.
>
>
>
> **4. Naming**
> Thank you for your suggestion regarding naming.
> In fact, we also had a hard time coming up with a suitable name for the method.
> We would consider alternative names for the proposed method.
>
> Again, we appreciate your effort and time in reviewing our paper. Please let us know if there are any further questions regarding our work.
>
> Best regards,
> Authors.
>
>
> Reference
>
> * Jiefeng Chen, Yixuan Li, Xi Wu, Yingyu Liang, Somesh Jha. ATOM: Robustifying Out-of-distribution Detection Using Outlier Mining. European Conference on Machine Learning (ECML) 2021.
> * Tom B. Brown, Dandelion Mané, Aurko Roy, Martín Abadi, Justin Gilmer. Adversarial Patch. ArXiv preprint arXiv:1712.09665. 2017.
> * Mahmood Sharif, Sruti Bhagavatula, Lujo Bauer, and Michael K. Reiter. A General Framework for Adversarial Examples with Objectives. ACM Transactions on Privacy and Security. 2019.
> * Matthias Hein, Maksym Andriushchenko, and Julian Bitterwolf. Why relu networks yield high confidence predictions far away from the training data and how to mitigate the problem. Computer Vision and Pattern Recognition (CVPR) 2019.

---

### Official Review · Reviewer_GYAe · 2021-11-01

**Correctness:** 4
**Technical Novelty And Significance:** 3
**Empirical Novelty And Significance:** 3
**Recommendation:** 6
**Confidence:** 3

**Main Review:**

## Pros
- This work studies an important problem with OOD detection, namely that the arbitrary choices of test datasets may lead to biased/overoptimistic estimates of detection quality. The paper is well written and easy to follow.
- The idea of extending search space of outliers by using a learned autoencoder makes intuitive sense, and the proposed solution appears to address most issues with existing evaluation protocol.
- Experiments are conducted with a large number of OOD detection methods, with very interesting results revealing the vulnerability of state-of-the-art detectors whose performance saturate on standard benchmarks.

## Cons / Questions
- As the authors have acknowledged in the paper, the proposed evaluation method does not fully eliminate the need of selecting OOD test sets, which significantly affects evaluation outcome (ranking of OOD detectors). That is, while the autoencoder learns to *interpolate* between outlier samples, it does not *extrapolate* beyond the chosen dataset, and therefore cannot provide full coverage of outlier distribution. Therefore, the results would be stronger if 1) repeated with more pairs of inlier-outlier datasets, and 2) trained on the concatenation of multiple outlier datasets.
- If the major concern is the coverage of OOD data used in existing evaluation protocols, can using a larger-scale test set (e.g. ImageNet) address the problem?
- As seen in qualitative examples of Figure 3, 4, 7-12, the majority of samples drawn from the adversarial distribution diverge from the space of natural images (especially when learned on SVHN). This makes the proposed approach closer to a benchmark of adversarial robustness, where images can be manipulated adversarially to fool a target classifier, especially when the latent dimension $D_z$ of the autoencoder is large. Does this make the benchmark unnecessarily difficult, when OOD detectors are expected to differentiate between *real* images? Can the authors further comment on the difference between the proposed task and adversarial robustness? Should one expect training OOD detectors with adversarial defense methods to improve performance on the proposed benchmark?

## Minor comments
- It would help to highlight best performing OOD detectors under each criterion in table 1.

**Summary Of The Paper:**

The paper proposes a novel evaluation framework for out-of-distribution (OOD) detection under worst-case scenarios. While existing benchmarks use real samples from datasets outside the training distribution, the authors propose instead to learn an adversarial outlier distribution against OOD detectors using an autoencoder model, with an auxiliary binary classifier that filters out inlier samples. Empirical experiments on CIFAR-10 (inlier) and SVHN/CelebA (outlier) datasets show that standard OOD benchmarks tend to produce overoptimistic results, and that prior methods with similar scores on standard benchmarks have diverging performance outside the predefined OOD test sets.

**Summary Of The Review:**

This paper presents a very interesting approach towards an objective and unbiased evaluation of OOD detection. While the proposed approach does not fully address the problems of existing evaluation protocols it identifies, I find the solution to be novel and intuitive, and the empirical results to be worthy for the community. Overall I am leaning towards accepting the paper, and would appreciate if the authors can address the questions enumerated in the Cons section above during rebuttal period.

---

> ### Author Response · Authors · 2021-11-23
> **Response for Reviewer GYAe**
>
>
> Dear Reviewer GYAe,
>
> We appreciate your constructive comments and are happy to reply to your questions.
>
> **1. Choice and Coverage of test OOD set**
>
>
> Evaluating OOD detectors with the full coverage of outlier space is generally not possible. It is not only because the outlier space is large but also because the true boundary of an inlier distribution is unknown.
> Using a larger test OOD dataset (e.g. ImageNet) would certainly raise the difficulty and fidelity of the evaluation result, but is still not sufficient to provide conclusive evidence that an OOD detector being evaluated is a perfect, ideal detector.
> Instead of aiming to achieve full coverage, our method focuses on the previously overlooked space of outliers.
>
> Suggestions for experiments are appreciated. More inlier-outlier pairs and concatenated outlier datasets would certainly strengthen the augment of the paper. We actually have conducted such experiments during our preliminary research.
> We will append the results to the appendix after cleaning up.
>
>
> **2. Connection to Adversarial Robustness**
>
>
> There is an intimate connection between adversarial robustness and OOD detection.
> OOD detectors need to detect adversarial samples, particularly adversarially perturbed outliers.
> As an outlier will still reside outside the inlier distribution after an attack, the attacked outlier remains as OOD, according to the definition of outlier provided in Section 2.1.
> An ideal OOD detector is supposed to classify any sample residing outside the inlier distribution's support as an outlier, including such adversarial outliers. In fact, some existing OOD detectors, such as ACET (Hein et al., 2019), GOOD (Bitterwolf et al., 2020), and CCU (Meinke and Hein, 2020), are built to detect adversarially perturbed outliers.
> However, existing state-of-the-art OOD detectors deviate significantly from such an ideal detector by failing to correctly classify outliers generated from adversarial distributions.
> We shall update the manuscript with a discussion on the connection to adversarial robustness.
>
> Furthermore, from the adversarial attack perspective, a threat model, i.e., the set of possible attacks, does not need to be confined in a small ball. There are a number of adversarial attack algorithms, including adversarial patch (Brown et al., 2017) and adversarial eye glasses (Sharif et al., 2018), that implement threat models different from norm balls.
> However, there is little previous work that investigates threat models beyond norm balls in the OOD detection setting.
> As an ideally robust OOD detector should be robust against any threat model, we believe that exploring attacks beyond the ball can be a meaningful contribution to the community.
>
> In fact, ACET and GOOD are outlier detectors that are trained with adversarial defense methods. However, assuming the norm-ball threat model, they show limited detection performance with respect to samples from adversarial distributions.
>
> **3. Regarding Other Comments**
> * We will mark the best-performing models with bold-face numbers in Table 1.
>
>
> Again, we thank you very much for your time and effort in reviewing. Please let us know if there are any further questions.
>
> Best regards,
> Authors.
>
> **Reference**
>
>
> * Matthias Hein, Maksym Andriushchenko, and Julian Bitterwolf. Why relu networks yield high confidence predictions far away from the training data and how to mitigate the problem. Computer Vision and Pattern Recognition (CVPR) 2019.
> * Julian Bitterwolf, Alexander Meinke, and Matthias Hein. Certifiably adversarially robust detection of out-of-distribution data. Neural Information Processing Systems (NeurIPS) 2020.
> * Alexander Meinke, and Matthias Hein. Towards neural networks that provably know when they don't know. International Conference on Learning Representations (ICLR) 2020.

---

> > ### Comment · Reviewer_GYAe · 2021-11-28
> > **Response to Authors' Comment**
> >
> > I would like to thank the authors for the detailed response to all questions in the review, especially the clarification regarding adversarial robustness and relation to other approaches. To my understanding, this paper mainly differs from prior work (ACET, CCU, GOOD, ATOM etc.) in that it considers outliers obtained by resampling in the latent space of an autoencoder, instead of input space of pixel values. I think this is a potentially interesting setting since methods targeted for $\ell_\infty$ perturbations do not seem to transfer well to the proposed setting. However, it would help to include a more in-depth discussion on this in the paper, beyond the single paragraph in page 4. In particular, I would appreciate if the authors can elaborate further on the two questions below:
> >
> > 1) Is the search space $\cal T$ considered in this paper, being confined to the output space of decoder $g(\cdot)$, strictly larger than the norm-ball model of previous work? In other words, should we expect OOD detectors trained on the adversarial distribution to be robust to $\ell_\infty$ perturbations on outlier samples?
> > 2) What are the practical importance of both settings in OOD detection without an adversary? For instance, can the proposed adversarial distribution resemble natural images better than $\ell_\infty$ perturbations or parameterized corruptions?
> >
> > On a different note, the experimental results with ensemble models are quite intriguing and would benefit from a deeper analysis. Why is the ensemble unable to improve detection performance of individual models (MD/NAE/OE), given that adversarial distributions against one model would have been ineffective for others? Can the authors speculate on the possible causes of this discrepancy?

---

### Official Review · Reviewer_hrej · 2021-11-02

**Correctness:** 4
**Technical Novelty And Significance:** 3
**Empirical Novelty And Significance:** 3
**Recommendation:** 6
**Confidence:** 4

**Main Review:**

Strengths:
 - well written
 - Good job at describing the general problem and setup
 - Reasonable methodology for generating the adversarial distribution
 - Strong evaluation

Weaknesses:
 - I find the use of a normal auto-encoder to be a little strange, as sampling from that latent space has issues [1]. I would much prefer a VAE or GAN that enforces a prior distribution. The authors could comment on these draw-backs in the main text, but I find the example generated images to be convincing of sufficient "OODness"
 - The connection between adversarial search and distribution is unclear. The sentence "sampling from an adversarial
distribution becomes equivalent to the adversarial search..." is somewhat buried in the text. The full discussion of adversarial search/optimization is, in my view, tangential to the main arguments of the paper. The paper's clarity would be greatly enhanced if the term adversarial was used less frequently and the generative aspect of the work was focused on.

Other:
 - Do the authors have a link to source code / website for the proposed leaderboards? If so, they should be linked. If not, leaderboard should be move to discussion section as future work.
 - section 6 & 7 could be combined
 - table 1 bolding is a little non-intuitive (typically reserved for best performing methods), perhaps italicized would convey the same message?
 - How many instances were generated for the adversarial distribution? 1000? This could be clarified in the main text.


[1] Bengio et al. "Representation learning: A review and new perspectives" 2013

Post rebuttal: I am generally okay with the current paper as is. I would prefer a better generative model and a bit a of a rewrite for less focus on the adversarials; for these reasons, I will be keeping my score.

**Summary Of The Paper:**

This paper introduces a new method for measuring an image classifier's robustness against out of distribution data by using adversarial search/distributions. The paper conducts experiments on various recent SOTA OOD models, and finds traditional metrics don't capture the whole picture when it comes to OOD detection.

**Summary Of The Review:**

This paper provides an interesting angle for investigating/evaluating OOD detectors. The paper is well-justified, with a mostly reasonable setup, and would be of interest for broader OOD community as a new metric.

---

> ### Author Response · Authors · 2021-11-23
> **Response for Reviewer hrej**
>
>
> Dear Reviewer hrej,
>
> First of all, we thank you for your time and effort in reviewing. Here, we would like to answer your questions.
>
> **1. Use of Normal Autoencoders**
> We use a vanilla autoencoder mostly due to its simplicity, but the proposed method is completely compatible with VAE or GAN. We just wanted to avoid making multiple design choices related to the generator, such as which GAN variant to use.
> We expect using choosing a different generator would alter the overall character of samples from the adversarial distribution, as it alters the zero-inlier space.
> If time permits, we will append experiments using different generators in the updated appendix.
>
>
> **2. Issues in Sampling from the Latent Space**
> Could you elaborate on exactly what issues regarding the latent space sampling you are worried about?
> I have looked up Bengio et al., 2013 and found that Section 9.4 discusses challenges in sampling.
> The section states that the mixing of MCMC can be very slow in a high-dimensional space where probability mass is concentrated in a manifold.
> In fact, this is why we perform MCMC in a lower-dimensional latent space because a Markov chain can move between density modes more easily than a chain in the original high-dimensional space.
> If this is not what you are concerned about, please let us know.
>
>
> **3. Adversarial Search**
> We shall clarify the connection between adversarial search and adversarial distribution in the updated manuscript. We will also update the manuscript to emphasize the generative aspect of the framework.
>
> **4. Regarding Other Comments**
> * The public benchmark leaderboard is still in progress. We have moved the paragraph on the leaderboard to the discussion section.
> * We generate 1,000 samples per experiment. We shall emphasize the fact in the caption of Table 1. We will also mark only the best-performing method bold in Table 1.
>
>
>
> Again, we appreciate your effort and time in reviewing our paper. Please let us know if there are any further questions regarding our work.
>
> Best regards,
> Authors.

---

### Official Review · Reviewer_HyPw · 2021-11-04

**Correctness:** 2
**Technical Novelty And Significance:** 2
**Empirical Novelty And Significance:** 2
**Recommendation:** 3
**Confidence:** 3

**Details Of Ethics Concerns:**

Just to note, as the authors stated, the outlier exposure method (OE) is trained on the 80M tinyimages dataset, which is no longer available due to an ethical issue. However, this work just borrowed the pre-trained model and has never accessed the 80M tinyimages dataset, so I think it is okay. They are just reporting the performance of the pre-trained model.

**Main Review:**

- The proposed benchmark is reasonable in a different way, so I recommend to redirect the goal and rewrite the paper.

- To my knowledge, OOD detection is not supposed to be robust against adversarial attacks, though some OOD detection works showed that their method is robust against adversarial attacks, e.g., [Lee et al., 2018]. Rather, the OOD detection task aims at detecting natural but different distributions, while adversarial attacks are mostly artificial. The proposed benchmark evaluates if OOD detection methods are robust against adversarial attacks, so the purpose is different from OOD detection works.

- The Mahalanobis distance method (MD) [Lee et al., 2018] leveraged adversarial attacks for validation in some of their experiments, which is worth to note as a prior work with a similar idea. I think MD should exhibit a good performance, if the authors followed this validation strategy.

## Post rebuttal

I do not change my rating, as I still think the purpose of the proposed benchmark should be described in a different way.

I think the answer in **1. OOD Detection and Adversarial Robustness** cannot justify if evaluating adversarial robustness on OOD samples can be generalized to unexplored space of (natural) outliers. By adding adversarial perturbations to the evaluation framework, the objective becomes to evaluate the adversarial robustness, not the OOD detection performance. To my understanding, related works are about mathematically certifiable or provable adversarial robustness on OOD detection problem, which is not really aligned with this work. As Reviewer XMWK concerned, the authors could clarify the relationship between this work and related works. It is true that there are "adversarially perturbed outliers which can be confirmed as OOD from human eyes," but I think detecting such artificially generated samples is a different problem from the general OOD detection problem.

Here I summarize the logical flow in the abstract below, where the transition from 2 to 3 is not convincing to me.
1. Current evaluation protocols cover only a small fraction of all possible outliers.
2. We want to test a detector over a larger, unexplored space of outliers.
3. To this end, we evaluate with samples from its adversarial distribution.

**Summary Of The Paper:**

This paper proposes to evaluate out-of-distribution (OOD) detection methods on adversarial distribution to detect unexplored space of outliers.

**Summary Of The Review:**

The proposed benchmark would be worth in a different way: not for a universal OOD detection, but adversarially robust OOD detection.

---

> ### Author Response · Authors · 2021-11-23
> **Response for Reviewer HyPw**
>
>
> Dear Reviewer HyPw,
>
> We appreciate your comments. Here, we would like to clarify some points of the paper that could have been explained clearer.
>
>
>
> **1. OOD Detection and Adversarial Robustness**
> Since adversarial robustness is most extensively investigated under the supervised learning setting, such as classification, it is possible to perceive that adversarial robustness is unrelated to OOD detection.
> Nonetheless, the adversarial perturbation can also be applied to an outlier image so that the perturbed image can fool an OOD detector while remaining as an outlier.
> Detecting such adversarially perturbed outliers has been recognized as a serious research problem for OOD detection in several previous works (Hein et al., 2019; Meinke and Hein, 2020; Bitterwolf et al., 2020).
> Figure 1 of Bitterwolf et al., 2020 visually illustrates that an adversarially perturbed outlier would reside outside of the inlier region and should be detected as an outlier.
> Figures 2 and 3 of Hein et al., 2019 provide examples of such adversarially perturbed outliers which can be confirmed as OOD from human eyes.
> Following the previous works, our definition of outlier provided in Section 2.1 also characterizes adversarially perturbed outliers as OOD which should be detected by an OOD detector.
>
>
> Meanwhile, an adversarial distribution aims to extend the dataset on which OOD detectors are evaluated over what existing evaluation protocols do.
> Existing works only utilized a finite set of test OOD samples or test OOD samples perturbed within a small norm-ball.
> An adversarial distribution generates outliers that are relatively far from existing test OOD data, e.g., beyond small balls around them, revealing previously unexplored weaknesses of OOD detectors.
>
>
> **2. Robustness of Mahalanobis Distance Detector** Adversarial attacks discussed in the MD paper (Lee et al., 2018) are different from adversarial attacks on outliers and cannot be compared directly. In Lee et al., 2018, the adversarial attack is performed to fool a separate multi-class classifier, e.g., DenseNet and ResNet. Note that the MD detector is not attacked directly.  In contrast, the adversarial attacks discussed in our paper are applied directly to the OOD detector, by producing adversarially perturbed outliers. In fact, it has been reported in the literature (Chen et al., 2020) that MD is not adversarially robust against attacks on MD itself, and we think this point is relevant and worth mentioning.
>
>
>
> Again, we thank you for your effort and time in reviewing. Please feel free to ask further questions if anything in the paper or the above response is unclear.
>
> Best regards,
> Authors.
>
> Reference
> * Kimin Lee, Kibok Lee, Honglak Lee, and Jinwoo Shin. A simple unified framework for detecting out-of-distribution samples and adversarial attacks. Neural Information Processing Systems (NeurIPS) 2018.
> * Matthias Hein, Maksym Andriushchenko, and Julian Bitterwolf. Why relu networks yield high confidence predictions far away from the training data and how to mitigate the problem. Computer Vision and Pattern Recognition (CVPR) 2019.
> * Julian Bitterwolf, Alexander Meinke, and Matthias Hein. Certifiably adversarially robust detection of out-of-distribution data. Neural Information Processing Systems (NeurIPS) 2020.
> * Alexander Meinke, and Matthias Hein. Towards neural networks that provably know when they don't know. International Conference on Learning Representations (ICLR) 2020.

---

### Decision · Program_Chairs · 2022-01-20

**Decision:**

Reject

**Comment:**

This paper studies the general problem of out-of-distribution (OOD) detection, where the goal is to detect outliers (i.e., points not in the distribution of training data) in the sample. The paper introduces a methodology for measuring robustness by using adversarial search/distributions. Experimental evaluation indicates that traditional metrics fail to fully capture OOD detection. The reviewers' evaluations of this work were mixed. Overall, there was consensus about the importance of the problem. Moreover, some of the reviewers argued that the submission contains some interesting new ideas. On the other hand, concerns were raised regarding lacking comparison to prior work, potential overselling of the contributions, and several aspects of the experimental evaluation. At the end, there was not sufficient support for acceptance. In its current form, the work appears to be slightly below the acceptance threshold.